# Estimation of hydraulic conductivity functions in karst regions by particle swarm optimization with application to lake Vrana, Croatia

Vanja Travaš[1], Luka Zaharija[2], Davor Stipanić[3], and Siniša Družeta[4]

[1]University of Rijeka, Faculty of Civil Engineering, Radmile Matejčić 3, 51000 Rijeka, Croatia
[2]Hidromodeling Ltd, Radmile Matejčić 10, 51000 Rijeka, Croatia
[3]Ryan Hanley Ltd, 1 Galway Business Park, Dangan, Galway, H91 A3EF, Ireland
[4]University of Rijeka, Faculty of Engineering, Vukovarska 58, 51000 Rijeka, Croatia

**Correspondence:** Vanja Travaš (vanja.travas@uniri.hr)

**Abstract.** To examine the effectiveness of various technical solutions for minimizing the adverse effects of salt water intrusion in lake Vrana, Croatia, a reliable mathematical model for describing the exchange of fresh and salt water between the lake and its surroundings is needed. For this purpose, a system of two ordinary and nonlinear differential equation is used. The variable coefficients represents hydraulic conductivity functions that are used to quantify groundwater flow and should be appropriately estimated by relying on data obtained by *in situ* measurements. In the abstract space of all possible hydraulic conductivity functions, the method of particle swarm optimization was used to search for functions which will minimize the difference between the predicted (modeled) and realized (measured) water surface elevation in the lake through the time span of 6 years (which includes relevant hydrological extremes - droughts and floods). The associated procedure requires the parameterization of conductivity functions which will define the number of dimensions of the search space. Although the considered mass exchange processes are significantly nonlinear, and the parametrization of hydraulic conductivity functions can define a search space with relatively large number of dimensions (60 dimensions were used to estimate the hydraulic conductivity functions of Vrana lake), the presented example confirms the effectiveness of the proposed approach.

## 1 Introduction

Water management in karst areas near the sea usually requires modeling water quantity and quality under different hydrological conditions (Bakalowicz , 2005). Karst area water resources are endangered by global littoralisation processes and also by negative anthropogenic impacts through which the requirements for fresh water quantities are progressively increasing. At the same time, natural processes manifested through climate change also negatively affect karst water resources by: (i) raising sea levels (which can endanger fresh water quality), (ii) reducing precipitation (the basis of which such sources are fed), and (iii) rising air temperatures (increasing the amount of fresh water evaporation). All this obliges us to the protection of such water resources, which in part requires mathematical modeling of water flow in karst conduits. In this context, the attention in the paper is paid to a specific problem of estimating the unknown hydraulic conductivity functions by which the groundwater flow in karst conduits is computed in the framework of semi-distributed lumped karst models. The related computational framework is well established (Gunn , 1986; Bergström and Forsman , 1973) and in many cases successfully implemented

(Fiorillo , 2011; Gàrfias et al. , 2007; Fleury et al. , 2007). Moreover, the semi-distributed lumped karst models is particularly interesting in case of poorly characterized karst aquifers. By representing the karst aquifer as a finite number of interconnected reservoirs (also known as hydrological compartments), the flow through karts conduits is represented as a consequence of the difference of water levels in interconnected reservoirs. In order to quantify the achieved water flow, the related heterogeneous hydrogeological properties are usually homogenized and described by parameters dependent on flow characteristics such as water level. As a consequence, the hydrological data are not spatially distributed and the simplified karst aquifer description relies on model calibration.

If the continuous time series of relevant hydrological data is known in advance (which was the case for the considered research area), the application of the principle of conservation of mass is very attractive for modeling the time change of water level in karst region. In this case, the dynamic behavior of such systems is described by a nonlinear first order ordinary differential equation (ODE) with variable coefficients (or system of ODEs). The variable coefficients should be determinate from model calibration. Since the system is described by nonlinear ODEs, the calibration methods based on the assumption that the karst aquifer can be represented by a connected series of linear reservoirs cannot be used. In these cases the calibration of the semi-distributed lumped karts model can be very challenging. Although, in such circumstances the model calibration procedure usually begins with a naive trial and error approach (Rimmer and Salingar , 2006) which is effective only in rare cases. Namely, the model variables are usually very sensitive to changes in calibration functions (which are depended on model variables). As an alternative, several methods have been developed for this purpose. A commonly used approach is based on statistical and correlation time-series analyses of measured hydrological data related to karst aquifers (Dubois et al. , 2020). However, in cases of strongly nonlinear dependencies, it is inevitable to base the calibration process on some auto correction method which relies on an definition of efficiency like the Nash-Sutcliffe efficiency measure or some modifications thereof (Charlier et al. , 2012).

Regardless of the adopted approach, the success of model calibration can be depended on the number of parameters that are subject of calibration. Namely, for a semi-distributed lumped karst model in which the exchange of water between intercon-nected karst region is modeled by more than one ODE, a different values of calibration parameters can result in similar model prediction (Wheater et al. , 2022; Ye et al. , 1997). In other words, there may be multiple solutions (known as multimodality) which consequently leads to unreliability in the physical interpretation of model parameters (Beven , 2006). In order to reduce the influence of overparameterization and obtain a unique solution, the number of all possible solutions should be reduced by introducing additional constraint conditions imposed on calibration parameters. If no generic property can be defined for a particular calibration parameter, by which the constraint condition can be formulated, the additional constraint conditions are obtained through analysis on the relative relationships between the available hydrological time-series (which is often car-ried out by correlation and cross-correlation analyses). In other words, solving multimodal problems most often requires the application of an algorithm for pattern recognition in the available hydrological time-series. For this reason, it should not be surprising that artificial neural networks (Kurtulus and Razack , 2010; Hu et al. , 2008; Coppola et al. , 2003; Coulibaly et al. , 2001) and different machine learning methods (Wunsch et al. , 2022) have found their application in calibration of semi-distributed lumped karst models. However, the mentioned approaches are not that suitable in cases where the constraint

conditions are known in advance and are given in the form of mathematical inequalities (as was the case in this paper). In such

cases, it is opportune to treat the calibration of model parameters as an optimization problem (Beasley et al. , 2012) in which multimodality is commonly encountered. In such circumstances, the calibration of model parameters requires the definition of objective function that is depended on design variables (i.e. model parameters). Since the objective function is usually defined as a measure of difference between the considered predicted value and the one obtained by field measurements, the calibration of model parameters is reduced to its minimalization. For this purpose, the domain of the objective function is searched in an

iterative fashion. Unless a specific search (local search) of the domain of the objective function is expected, the parameters of a karst model can be effectively calibrated by genetic algorithms (Lu et al. , 2018). For more demanding optimization problems, in which it is expected that the objective function has many local minima permeated throughout the entire domain of the objective function (multimodality), both global and local search is necessary. In these situations, the bio-inspired algorithm known as particle swarm optimization (PSO method) is more suitable because it is based on simultaneous local and global search of

the domain of the objective function (Qian et al. , 2019) and so, it is very attractive for solving multimodal problems (Özcan and Yilmaz , 2007). Moreover, this approach has previously been successfully applied to calibrate groundwater flow models in alluvial aquifers (Haddad et al. , 2013; Mahmoud et al. , 2021) but also for calibrating flow parameters in environmental models (Zambrano-Bigiarini and Rojas , 2020). In order to examine the application of the PSO method and indicate the possibilities it offers in a contest of karst modeling, it was applied to the estimation of hydraulic conductivity functions used for modeling the

exchange of fresh and salt water in lake Vrana, Croatia.

## 2 Study area

The proposed procedure for estimating hydraulic conductivity functions has been successfully applied in quantifying water flow through the bed of lake Vrana (which is the largest natural lake in Croatia, with a water surface area of more than 30 square kilometers). Lake Vrana is a cryptodepression separated from the Adriatic sea by an narrow karst ridge (width varying

from 0.8 to 2.5 km) through which fresh and salt water can be exchanged. The exchange of water is bidirectional and the orientation of the established groundwater flow, which is important in the context of preserving fresh water quality, depends on achieved pressure gradient along karst conduits. In this regard, it is important to note that lake Vrana water level varied from 0.03 to 2.25 m a.s.l. in the time span from 1948 to 2010 (Rubinić and Katalinić , 2014). Although the lake water level was entire time above the sea level, the risk of salinization can be recognized by: (i) the fact that the average lake water level in the

specified period was 0.83 m a.s.l., and (ii) taking into account that the pressure gradient depends also on the relative deference in fresh and salt water density (so that the salt water intrusion is established even at equal water levels). Moreover, in August 2012, due to unfavorable hydrological trends (attributed to climate change), the depth of the lake water was only 30 cm and a very high salinity of as much as 17‰ was recorded. This was a consequance of site specifics, namely the very close proximity of the sea, and the relatively shallow water in the lake. Furthermore, it should be noted that the lake bed in the deepest point is

only 3 m below mean sea level and thus the lake is under constant danger of salinization (Rubinić , 2014).

In addition to the previously mentioned, it should be emphasized that the quality and quantity of water are also significantly affected by the presence of Prosika channel through which fresh and salt water can be exchanged by: (i) surface flow through the channel, and (ii) groundwater flow through the porous channel bed. Prosika channel was originally dug in 1770 to attain new agricultural areas near lake Vrana that needed protection from seasonal floodings. With a total length of 770 m and trapezoidal cross-section with a channel bottom width of 8 m, Prosika channel connects lake Vrana and the Adriatic sea. The location of lake Vrana, Prosika channel and Adriatic sea is shown in Fig. 1. Throughout history, Prosika channel has undergone several geometric adaptations which led to weir of fixed height and with a crest at 0.41 m a.s.l. on the downstream side of the channel (which has the role of separating fresh and salt water in case the sea is at a lower elevation). However, the fixed crest height, which was originally determined as a compromise between maximizing channel flow area during flooding and minimizing channel flow area during salt water intrusion, it is no longer adequate because in recent history the problem of salt water intrusion has repeatedly arisen.

In order to reduce the negative consequences of salinization (or significantly reduce salt water intrusion), it is necessary to intervene in the process of exchange of fresh and salt water in lake Vrana by some technical solution. For this purpose, the adopted technical solution should: (i) increase the storage time of fresh water in lake Vrana (which is supplied only through precipitation and surface and groundwater flow from karst aquifer), and (ii) reduce the intrusion of salt water from Adriatic sea (by surface water flow through Prosika channel or groundwater flow thought the porous Prosika channel bed and lake Vrana bed). Regardless of the adopted technical solution, the resulting effect must be quantified by comparing the volume of salt and fresh water in lake Vrana under different and relevant hydrological scenarios. For this purpose, it is necessary to formulate a mathematical model that can be used to simulate the exchange time of fresh and salt water in lake Vrana under different hydrological conditions, which in turn requires a realistic description of the groundwater flow (while the surface flow through Prosika channel can be modeled relatively easily). Thus, the modeling problem is reduced to defining the suitable hydraulic flow conditions in unknown network of conduits in the surrounding karst aquifer.

Within the basin of lake Vrana, few groups of rocks can be recognized (Rubinić , 2014). First of all, these are Upper Cretaceous limestones, i.e. very permeable rocks within which an underground hydrographic network has been developed. On the other hand, it is also possible to determine the area within which the dolomites and limestones of the lower part of the Upper Cretaceous alternate, forming a medium permeable layer that can slow down the flow of underground water. Finally, a large part of the basin consists of impermeable or very poorly impermeable flysch deposits that in some places cause the formation of surface flows. For calibrating the model parameters, these surface flow components will be set based on known *in situ* measurements. On the other hand, the groundwater flow components, which are realized as a consequence of the developed hydrographic network in the Upper Cretaceous limestones, will be modeled using the semi-distributed lumped karst model, relying on the assumption of a fully turbulent flow.

## 3 Research method

The semi-distributed model or pipe flow model (Gill et al. , 2021; Schmidt et al. , 2014; Bailly et al. , 2012; Thrailkill , 1974), was used to model storage dynamics of lake Vrana and its hydrogeological connectivity with: (i) surrounding karst basin (from which it is supplied with fresh water), and (ii) Adriatic sea (where fresh water from lake Vrana sinks). Accordingly, groundwater flow was modeled using the assumption of fully turbulent and partially saturated water flow through karst conduits in the phreatic and epihreatic zones (Shoemaker et al. , 2008; Bonacci , 1993), neglecting the Darcy's flow component. Under these assumptions, the karst conduit networks can be conceptualized as a system of connected pipelines so that the relevant hydraulic parameters are related through the Darcy-Weisbach equation by introducing hydraulic conductivity functions.

### 3.1 Hydraulic conductivity function

For an ideal conduit with circular cross-section, the Darcy-Weisbach equation can be written as

$$h_a - h_b = \lambda \frac{L}{D} \frac{\overline{v}^2}{2g} \tag{1}$$

where $h_a$ and $h_b$ represents pressure heads [L] at the opposite ends $a$ and $b$ of the conduit, $\lambda$ the Darcy's friction coefficient [1], $L$ the length of the conduit [L], $D$ the diameter of the conduit [L], $\overline{v}$ the average flow velocity [LT$^{-1}$] and $g$ the acceleration of gravity [LT$^{-2}$].

The flow rate $q_{ab}$ thorough the conduit can be obtained by multiplying the cross-section area $A$ with the average flow velocity $\overline{v}$ from Eq. (1), thus introducing the flow model

$$q_{ab} = \underbrace{A\sqrt{\frac{2gD}{\lambda L}}}_{c_{ab}(\Delta h_{ab})} \sqrt{h_a - h_b} \tag{2}$$

which can be generalized to generic flow conditions. Namely, the flow rate $q_{ab}$ through the conduit can be related to the square root of a pressure head difference $\Delta h_{ab} = h_a - h_b$ on the right hand side by a proportionality factor $c_{ab}$ [L$^{5/2}$T$^{-1}$] which describes the combined influence of the geometric and kinematic properties of the flow. Moreover, since both flow characteristic will be affected by the pressure head difference, $c_{ab}$ can be interpreted as a hydraulic conductivity function with the argument $\Delta h_{ab}$. To generalize the flow model in respect to the flow direction, which can change depending on the sign of the pressure head difference $\Delta h_{ab}$, Eq. (2) can be rewritten as

$$q_{ab} = sgn\, \Delta h_{ab} \cdot c_{ab}(\Delta h_{ab}) \cdot \sqrt{|h_a - h_b|} \tag{3}$$

In order to include the dependence of a pressure gradient on the difference in water density at the conduit ends, the pressure head $h_b$ in Eq. (3) should be modified by a factor $r_\rho$ that represents the ratio between the density of water $\rho_b$ with pressure head $h_b$ and density of water $\rho_a$ with pressure head $h_a$, which leads to the flow model

$$q_{ab} = sgn\, \Delta h_{ab} \cdot c_{ab}(\Delta h_{ab}) \cdot \sqrt{|h_a - r_\rho h_b|} \tag{4}$$

where the hydraulic conductivity function $c_{ab}(\Delta h_{ab})$ should be estimated by inverse modeling (Li et al. , 2018; Nematolahi et al. , 2018), relying on available data obtained from *in situ* measurements. It should be pointed out that the functions in consideration can be highly nonlinear due to the nonlinear effect of friction (for flow in pipes defined by the Colebrook equation), and even more due to the change in the geometry of the conduit network that can vary as a function of surface water and groundwater level.

## 3.2 Conceptual model

The fresh water and seawater exchange between lake Vrana and Adriatic sea, as well as the exchange of fresh between lake Vrana and its surrounding karst aquifer, can be described by Eq. (4). For this purpose, the mathematical model must include three variables: (i) sea level $h_s$, (ii) lake water level $h_l$, and (iii) karst groundwater level $h_k$. Within a given time domain, the change of sea level $h_s$, described by function $h_s(t)$ over time $t$, is given in advance and the functions $h_l(t)$ and $h_k(t)$ will be treated as unknown quantities that will be approximated for given initial and hydrological conditions. The corresponding semi-distributed lumped karst model will result in a system of three interconnected reservoirs that are introduced to conceptually represent the hydrological compartments: (i) Adriatic sea, (ii) lake Vrana, and (iii) karst aquifer. The relative relationship between the introduced hydrological compartments, with the corresponding degrees of freedom $h_s$, $h_l$ and $h_k$, and theirs interconnected flow components are illustrated in Fig. 2.

Apart from the groundwater flow components, the conceptual model should also include the exchange of water between the introduced hydrological compartments achieved by surface water flow. Namely, the surface water component plays an important role because it feeds lake Vrana with fresh water (like the groundwater flow component from the karst aquifer). On the other hand, the surface flow component between lake Vrana and the Adriatic sea, achieved through Prosika channel, can adversely affects the quantity and quality of water in lake Vrana. Namely, in case of downstream flow (from the lake towards the sea), the quantity of fresh water in lake Vrana is reduced, increasing the relative difference of variables $h_s$ and $h_l$ in the unfavorable direction. Otherwise, i.e. in case of upstream flow (from the sea towards the lake), the percentage of salt water in the lake will increase. These situations, as well as the case in which there is no surface flow through Prosika channel, will depend on the boundary conditions at the channel ends, which should be included in the corresponding mathematical model.

## 3.3 Mathematical model

To formulate a formal mathematical representation of the previous conceptual model, a principle of mass conservation can be applied on reservoirs introduced to model lake water level $h_l$ and groundwater level $h_k$ in the surrounding karst aquifer. Accordingly, a principle of mass conservation for groundwater in the karst aquifer requires

$$A_k(h_k)\frac{dh_k}{dt} = \underbrace{q_{k,pr}}_{\text{given}} - \underbrace{\Delta h_{kl} \cdot c_{kl}(\Delta h_{kl}) \cdot \sqrt{|h_k - h_l|}}_{q_{kl,gw}} \tag{5}$$

where the function $A_k(h_h)$ relates the groundwater level $h_k$ [L] to corresponding horizontal-section area $A_k$ [L$^2$] of karst conduit networks, $q_{k,pr}$ represents the fresh water inflow from precipitation [L$^3$T$^{-1}$] and $q_{kl,qw}$ the groundwater flow component

between karst conduit networks and lake Vrana, calculated by Eq. (4) where $c_{kl}(\Delta h_{kl})$ denotes the corresponding hydraulic conductivity function with the argument defined as $\Delta h_{kl} = h_k - h_l$. It should be highlighted that the function $A_k(h_h)$ denotes the arrangement of the cross-section surface areas of cracks openings in the karst environment with respect to groundwater level and thus includes the surfaces of karst conductors, caves, caverns, etc. Like hydraulic conductivity functions, this function is also unknown in advance and should be obtained by model calibration respecting the characteristic of a progressive decrease with the rise of groundwater level (due to the dissolution process that creates larger openings in the deeper parts of the karst).

Similarly, the principle of mass conservation for lake Vrana requires

$$A_l(h_l)\frac{dh_l}{dt} = \underbrace{q_{kl,sw} + q_{l,pr} - q_{l,ir} - q_{l,ev}}_{\text{given}} + q_{kl,gw} - q_{cs,sw} \underbrace{-\Delta h_{cs} \cdot c_{cs}(\Delta h_{cs}) \cdot \sqrt{|h_c - r_\rho h_s|}}_{q_{cs,gw}} \underbrace{-\Delta h_{ls} \cdot c_{ls}(\Delta h_{ls}) \cdot \sqrt{|h_l - r_\rho h_s|}}_{q_{ls,gw}}$$

(6)

where the function $A_l(h_l)$ relates lake water level $h_l$ [L] to corresponding water surface area $A_l$ [L$^2$] in lake Vrana (known by *in situ* measurements), $q_{kl,sw}$ represents the surface water inflow [L$^3$T$^{-1}$], $q_{l,pr}$ the fresh water inflow from precipitation on water surface area $A_l$ [L$^3$T$^{-1}$], $q_{l,ir}$ the fresh water outflow from irrigation [L$^3$T$^{-1}$], $q_{l,ev}$ the fresh water outflow from evaporation [L$^3$T$^{-1}$], $q_{cs,sw}$ and $q_{cs,gw}$ are the surface water and groundwater flow components achieved between Prosika channel and the Adriatic sea [L$^3$T$^{-1}$] and $q_{ls,gw}$ is the groundwater flow component between lake Vrana and Adriatic sea [L$^3$T$^{-1}$]. It should be noted that the first four terms on the right hand side are known from *in situ* measurements.

As in Eq. (5), the groundwater flow components in this case are computed by Eq. (4) so that $c_{cs}(\Delta h_{cs})$ and $c_{ls}(\Delta h_{ls})$ represent hydraulic conductivity functions for groundwater flow components achieved between: (i) Prosika channel and Adriatic sea, and (ii) lake Vrana and Adriatic sea. However, it is opportune to note that the argument $\Delta h_{ls} = h_l - h_s$ can be explicitly computed and that the argument $\Delta h_{cs}$, which represents the difference in water level in Prosika channel $h_c$ and sea level $h_s$, requires a modeling procedure through which the water surface profile $h_s(s)$ along the channel station $s$ is computed for a given set of boundary condition. For this purpose, a standard step method was used. Moreover, it should be note that the sinking flow component along the channel, introduced in Eq. (6) by the term $q_{cs,gw}$, requires the hydraulic conductivity function $c_{cs}(\Delta h_{cs})$ which was set to be linear so that the difference in water level $\Delta h_{cs} = 0.8$ corresponds to the value of $c_{cs} = 0.5$ m$^{5/2}$s$^{-2}$ (as established by *in situ* measurements).

By the principle of mass conservation, Eq. (5) and Eq. (6) form a system of two nonlinear ordinary differential equations that can be used to describe the relationship between quantities $h_k$ and $h_s$ for a given set of initial conditions and known hydrological parameters over the considered time domain. However, to obtain reliable solutions, there are three functions that should be obtained by inverse modeling, relying on known data obtained by *in situ* measurements: (i) $c_{kl}(\Delta h_{kl})$, (ii) $c_{ls}(\Delta h_{ls})$, and (iii) $A_k(h_h)$. However, as the functional relationship between groundwater flow $q$ and pressure gradient is nonlinear, as given by Eq. (3) and Eq. (4), it should be emphasized that even a small change in one of the estimated hydraulic conductivity functions will result in a relatively large difference in the predicted function $h_l(t)$. Moreover, the deviation between the predicted and realized lake water level, at any point in time, can be related to the volume of water that is transferred to the rest of the time domain, so deviations between predicted and measured lake water level can only increase over time (because the mathematical

model is based on the principle of mass conservation). For this reason, the problem of estimating the hydraulic conductivity functions is usually very complex (especially in large time domains needed to take into account different hydrological situations).

## 3.4 Particle swarm optimization

The unknown functions are estimated by model calibration (Kuok and Chiu , 2012; Zambrano-Bigiarini and Rojas , 2013), which is performed in an iterative fashion, relaying on available hydrological data. The procedure requires a representative time domain (with hydrological extremes) in which all the relevant data are well documented by *in situ* measurements. In that case, the mathematical model can be used to predict the function $h_l(t)$ under the same hydrological conditions that led to the change in lake water level $\hat{h}_l$ observed by *in situ* measurements and described by the function $\hat{h}_l(t)$. In that case, the assumed functions $c_{kl}(\Delta h_{kl})$, $c_{ls}(\Delta h_{ls})$ and $A_k(h_h)$, which can be interpreted as design functions, are validated by comparing the predicted $h_l(t)$ and achieved $\hat{h}_l(t)$ change in lake water level $h_l$. The difference in consideration can be measured by a function $G$ defined as a sum of squared differences $h_l(t_n) - \hat{h}_l(t_n)$ performed over a finite number of points $t_n$ in the given time domain (where $n$ ranges from 0 to $n_{\Delta t}$). Accordingly, the shaping of the design functions can be viewed as an optimization problem that requires the minimization of function $G$ which in that case represents the objective function.

Since the optimization procedure in the proposed methodology is conducted numerically (not analytically), the design functions are represented by a series of function values equidistantly distributed between the minimal and maximal values of each design function domain. These discrete values, used to approximate the design functions, are collected in a vector $\mathbf{x}^{(e)}$ and updated after each evaluation step $(e)$.

In recent years, modern stochastic global optimization methods have been successfully applied in many difficult real world problems. One such optimization method is particle swarm optimization (PSO), which has been employed in several hydrological modeling problems. In PSO method (Clerc , 2010), the search space of all possible design functions is explored by $n_p$ agents called particles, each with their own iteratively updated set of design variables $\mathbf{x}_p^{(e)}$.

In accordance with the above, particle $p$ in an evaluation step $(e)$ evaluates its design vector $\mathbf{x}_p^{(e)}$ by the use of the objective function

$$G(\mathbf{x}) = \sum_{n=0}^{n_{\Delta t}} \left( h_l(\mathbf{x}, t_n) - \hat{h}_l(t_n) \right)^2 \tag{7}$$

where the function $h_l(\mathbf{x}, t_n)$ represent the model prediction for design variables

$$
\mathbf{x}_p^{(e)} =
\left\{
\begin{array}{c}
c_{kl}\left(\Delta h_{kl,min}\right) \\
\vdots \\
c_{kl}\left(\Delta h_{kl,max}\right) \\
c_{ls}\left(\Delta h_{ls,min}\right) \\
\vdots \\
c_{ls}\left(\Delta h_{ls,max}\right) \\
A_k\left(h_{k,min}\right) \\
\vdots \\
A_k\left(h_{k,max}\right)
\end{array}
\right\}
\tag{8}
$$

After evaluating all vectors $\mathbf{x}_p^{(e)}$ in the current evaluation step $(e)$, the vectors $\mathbf{x}_p^{(e+1)}$ in the next evaluation step $(e+1)$ are computed by kinematic analogy

$$
\mathbf{x}_p^{(e+1)} = \mathbf{x}_p^{(e)} + \mathbf{v}_p^{(e+1)}
\tag{9}
$$

where $\mathbf{v}_p$ can be interpreted as velocity vector of particle $p$.

The crucial element of PSO is related to the computation of the velocity vector $\mathbf{v}_p^{(e+1)}$, which is inspired by the movements of swarms in a collective search of some biological need (e.g. food). For this purpose, particle movement is affected by three components: (i) inertial component which describes the tendency to preserve the current direction and speed of motion, (ii) a component of self-confidence that describes the tendency to explore the search space on the basis on personal search experience (particle memory influence), and (iii) a component of collective-influence that describes the attraction of the very best solution found among the members of the swarm informing the particle in question (swarm memory influence). This collective-influence is most often implemented as the influence of the information of the best solution found by the entire swarm, i.e. for the purpose of information sharing, the swarm is understood to be operating as a fully connected graph. Therefore, the velocity vector $\mathbf{v}_p^{(e)}$ for each particle $p$ can be updated according to the given description that can be mathematically represented by

$$
\mathbf{v}_p^{(e+1)} = \underbrace{w \cdot \mathbf{v}_p^{(e)}}_{\text{inertia}} + \underbrace{c_1 \cdot \mathbf{r}_1 \circ \left(\mathbf{x}_{p,best}^{(e)} - \mathbf{x}_p^{(e)}\right)}_{\text{particle memory influence}} + \underbrace{c_2 \cdot \mathbf{r}_2 \circ \left(\mathbf{x}_{g,best}^{(e)} - \mathbf{x}_p^{(e)}\right)}_{\text{swarm memory influence}}
\tag{10}
$$

where $w$ is the dimensionless inertia parameter, $c_1$ and $c_2$ are dimensionless parameters used to describe the relative importance between the influence of particle memory and swarm memory, respectively, $\mathbf{r}_1$ and $\mathbf{r}_2$ are random vectors with components taken from an uniform statistical distribution between 0 and 1 and introduced to replicate the stochastic components of particles movement, $\mathbf{x}_{p,best}^{(e)}$ is the vector of design variables in a history of particle $p$ by which the objective function $G$ reach the minimal value (local optimum vector), $\mathbf{x}_{g,best}^{(e)}$ is the vector of design variables extracted from the search history of all particles by which the objective function reach the so far established minimum (global optimum vector) and $\circ$ denotes the Hadamard product. After each particle evaluation, a check for updating vectors $\mathbf{x}_{p,best}^{(e)}$ and $\mathbf{x}_{g,best}^{(e)}$ is performed. From the assumed initial position

$\mathbf{x}_p^{(0)}$ and velocity $\mathbf{v}_p^{(0)}$ for all particles $p$, the optimization algorithm given by Eq. (9) and Eq. (10) is repeated in an iterative fashion until the objective function $G$, for design variables $\mathbf{x}_p^{(e)}$, yields value lower than some predefined convergence limit, or some other stopping criteria is achieved.

It should be noted that for a problem in consideration there are some constrains that can be superposed to the unknown functions $c_{kl}(\Delta h_{kl})$, $c_{ls}(\Delta h_{ls})$ and $A_k(h_h)$, by reducing the search space and consequently increasing the efficiency of the optimization algorithm. Namely, as a consequence of the relation between pressure gradient and discharge, the values of functions $c_{kl}(\Delta h_{kl})$ and $c_{ls}(\Delta h_{ls})$ must increase as $\Delta h_{kl}$ and $\Delta h_{ls}$ increase. In other words, functions $c_{kl}(\Delta h_{kl})$ and $c_{ls}(\Delta h_{ls})$ are monotonically increasing functions. On the other hand, a similar condition can be applied to function $A_k(h_h)$. Namely, by limestone dissolution, underground water currents and loads from upper karst deposits, it is reasonable to expect that the aquifer in consideration contains caves and caverns. Also, it is usually reasonable to assume that the volume of caves and caverns increase with the aquifer depth so that the condition of monotonicity in growth of $A_k(h_h)$ can be used for each of $n$ points used to represent the corresponding function values.

## 4  Results

The presented methodology was applied for the estimation of design functions $c_{kl}(\Delta h_{kl})$, $c_{ls}(\Delta h_{ls})$ and $A_k(h_h)$, present in the previously presented mathematical model given by Eq. (5) and Eq. (6) and developed for modeling the fresh and salt water exchange in lake Vrana. Since the resulting computational algorithm is based on an iterative procedure by which the calibration of design functions is performed, it is appropriate to calibrate them for a relatively long and representative time domain within which hydrological extremes are present. For this purpose, the time domain from the beginning of 2010 to the end of 2015 was chosen to calibrate the subject design functions. Namely, this time domain contains the previously mentioned case of extremely low lake water levels, but also several cases of flood waves. These extreme events can be recognize in Fig. 3 showing the measured lake water level and sea level. The variability of hydrological conditions is necessary to reduce the multimodality of the optimization problem, i.e. to reduce the search space of design functions that must be uniquely defined and ensure the agreement of modeled and measured lake water levels for dry and wet periods. The calibration process requires that in the selected time domain all the relevant hydrological and other data, present in the corresponding mathematical model (terms on the right hand side of the differential equations), must be known by *in situ* measurements (as is known in the case of lake Vrana).

From the computational point of view, it should be noted that such large time domains require stable numerical integration and therefore the system of ordinary differential equations given by Eq. (5) and Eq. (6) was solved by applying an implicit numerical scheme. On the other hand, it should also be noted that there are physical circumstances that can influence the choice of time step. Namely, in the present case the size of the time step is conditioned by sea level dynamics illustrated in Fig. 3. In other words, the estimation of design functions must be carried out to take into account changes in groundwater flows that occur between lake Vrana and Adriatic sea (same as between Prosika channel and Adriatic sea) within one day as a result of changes in sea level. For this purpose, groundwater flow components are determined on the basis of hourly changes in sea

level, while the lake water level does not change significantly within one day. The resulting numerical scheme is implemented into a computational algorithm written in Python. The initial conditions were given by the model variables $h_l(t_0)$ and $h_k(t_0)$ defined at time $t_0$ i.e. at the beginning of the time domain. The initial condition $h_l(t_0)$ was set to 0.81 m a.s.l. which is known by field measurements (as can be recognized in Fig. 3). On the other hand, the variable $h_k(t_0)$ was set to 2.2 m a.s.l. and determinate from model calibration so that a relatively rapid raise in water level $h_l$, at the beginning of the time domain, is obtain (as evidenced by *in situ* measurements shown in Fig. 3).

As explained above, PSO was used to estimate the considered design functions. For this purpose, each of the three design functions is discretized with 20 points so that the search space is defined with 60 dimensions. The values of design functions in this points will change during iterations, but it is also good to recognize that the design function domains will also vary during iterations. Namely, each examined case of design functions will lead to different functions $h_l(t)$ and $h_k(t)$ and thus to different domain of independent variables $\Delta h_{kl}$ and $\Delta h_{ls}$. By applying the above mentioned criteria that the considered design functions must meet, 22 evaluation steps conducted with 50 particles were required to reach an acceptable error of the predicted lake water level when compared to the field measurements, as shown in Fig. 4. The convergence of the optimization process is illustrated in figure 5 which shows the value of the objective function at points $\mathbf{x}_{g,best}^{(e)}$ of the global optimum respect to the iteration number. For the adopted parametrization of the calibration functions, the objective function reached the lowest possible value, and further reduction of its value would require a larger number of parameters, i.e. a denser discretization of the calibration functions (more than 20 point per functions). On the other hand, such a procedure would significantly affect the number of necessary iterations to reach a smaller error, as well as the number of required particles (because the search space would be larger). In this sense, the parameterization of the calibration functions is determined based on a compromise between computational efficiency and acceptable minimum value of the objective function.

In order to compare the presented approach with other approaches, it should be emphasized that the framework of the model is defined by a system of two ordinary and non-linear differential equations with variable coefficients that also define the calibration functions. In these circumstances, automated methods such as genetic algorithms are usually used (Lu et al. , 2018; Nematolahi et al. , 2018). On the other hand, if the trial and error method shows that the results are very sensitive to the calibration functions (model parameters), the application of genetic algorithms is probably not the most appropriate. Namely, this sensitivity of model results to calibration functions indicates a large number of local minima of the objective function by which the multimodality of the problem can be recognized. In such circumstances, it is not only necessary to carry out a global search of the domain of the objective function, as carried out by the method of genetic algorithms, but it is also necessary to examine local minima, that is, to enable a more detailed search of individual parts of the domain by carrying out local searches. Moreover, the search for local minima must be adaptive so that the solution in the current iteration can be updated by the new local solution that results in a more favorable variant of the calibration parameters. In this way, the possibility of searching for a larger number of local minima is realized, which is necessary for such multimodal problems. Considering all of the mentioned, and by noting that the model in question showed the characteristics of multimodal problems, the calibration of the model was performed using the PSO method which simultaneously performs global and local search of the domain of the objective function (Özcan and Yilmaz , 2007; Kuok and Chiu , 2012; Zambrano-Bigiarini and Rojas , 2013). Considering the

experience gained from the performed analysis, the application of the PSO method can be recommended for the calibration of a semi-distributed lumped karst models based on a system of nonlinear ODEs. Although, the calibration procedure should be carried out taking into account the uncertainty in the input data of the model as shown below.

## 4.1 Uncertainty in measurements and calibration functions

Since the model refers to a relatively large karst area, and relies on a large database obtained by field measurements over the time span of 6 years, the model calibration procedure should be conducted by considering the uncertainty of the input data. For this purpose, it should be noted that the flow component $q_{kl,sf}$, which is related to surface water flow into lake Vrana, has a relatively low degree of uncertainty because it was obtained by continuous monitoring of surface water level in all tributaries to the lake (relating water level to flow rate under the assumption of stationary flow conditions). Similarly, the flow component $q_{l,ir}$, related to water extraction from lake Vrana for irrigation of agricultural land, is also quite reliable data since it refers to the volume of water that is charged. On the other hand, the flow component $q_{l,ev}$, related to evaporation from the water surface of lake Vrana, contains a certain degree of uncertainty, but again much smaller than the main source of uncertainty which is related to the flow components $q_{l,pr}$ and $q_{k,pr}$ i.e. precipitations on the water surface of the lake and the surrounding karst aquifer. In order to take into account these uncertainties, the error in the flow components $q_{l,pr}$ and $q_{k,pr}$ were modeled stochastically by Gaussian distribution $\mathcal{N}(\mu, \sigma)$ in which $\mu$ is the mean value and $\sigma$ is the related standard deviation. Accordingly, for each day $i$ in the considered time domain, the input data of precipitation were obtained by adding the error value randomly sampled from the statistical distribution $\mathcal{N}(p_i, \sigma)$ to a known measured data $p_i$ of precipitation (with a standard deviation $\sigma$ of 5 mm/day). Therefore, the calibration of the model was carried out for a larger number of cases of precipitations (flow components $q_{l,pr}$ and $q_{k,pr}$), which included the uncertainty of the precipitations data and enabled the examination of its influence on the unknown calibration functions, i.e. describe the range of potential hydraulic conductivity functions at some probability level. In other words, the calibration of the model was conducted in an iterative fashion by repetitive application of the previously described procedure (without changing the initial parameters of the PSO method). For each generated case of precipitation, the PSO method was applied to estimate the required values of the calibration functions. In order to evaluate the estimated functions equally, for all different cases of precipitations, the application of the PSO method was carried out until the global optimum did not fall below the predefined tolerance. In this way, the calibration error in all cases was of the same order of magnitude.

In accordance with the above description, the calibration of the unknown functions was carried out for 200 cases of precipitations. In this way, for each discrete coordinate in the domain of the unknown functions, a set of 200 possible function values was estimated. In other words, the values of the calibration functions are represented with a statistical distribution for each discrete domain coordinates. In order to graphically display the schedule of obtained values, the so-called box plot diagram was used by showing the five number summary of the obtained data set (for each point in the function domain): minimum, lower quartile, median (illustrated by an orange line), upper quartile and maximum. The results of the described calibration process are shown in Fig. 6, 7, and 8. For the calibration function $c_{kl}(\Delta h_{kl})$, it should be noted that in the area of the domain from 0 to 2.5 m a.s.l. there is very little scatter around the obtained mean value (blue dots connected by a blue line) and after that a slightly larger scatter can be recognized in the central part of the domain (Fig. 6), but again the trend of the function

is well articulated (with a mean value near by the middle of the interquartile range). In the case of the calibration function $c_{ls}(\Delta h_{ls})$, the deviations of the obtained values around the mean values of the hydraulic conductivity (blue dots) are almost
constant (Fig. 7). In contrast to the previous hydraulic conductivity function, in this case the model predicts the exchange of water in both directions, i.e. from the lake to the sea and vice versa. Scattering around the mean value of the function $A_k(h_k)$ is not uniform within the domain and the maximum uncertainty is contained in the middle of the domain i.e. in the range from 3.0 to 5.5 m a.s.l. (Fig. 8).

In order to reduce the uncertainty of the calibration functions, additional conditions can be included within the presented
calibration procedure. Namely, the allowed ranges of values of the calibration functions can be specified within the optimization procedure for a particular part of the domain. For this purpose, it should be noted that the most sensitive areas of the domain of calibration functions can be determined by uncertainty analysis (as in the case of lake Vrana) and thus define the scope of additional field measurements that will provide the necessary data by which additional conditions for calibration functions can be prescribe (which will consequently reduce uncertainty). In order to carry out the analysis of the dynamics of the exchange
of fresh and salt water in lake Vrana (under different protection conditions), the calibration functions shown by the blue line in Fig. 6, 7 and 8 were used (defined by joining the mean values of the obtained statistical distributions).

## 4.2    Analysis of the existing protection against seawater intrusion

The calibrated mathematical model was used to conduct a more detailed analysis of fresh and salt water exchange in lake Vrana during the considered time domain. Namely, the total volume of fresh or salt water that fills or empties lake Vrana in a unit of
time can be decomposed into its constituent parts given by the corresponding terms in Eq. (5) and Eq. (6). Accordingly, Fig. 9 illustrates the origin of water volumes entering or exiting from lake Vrana, where positive volumes denotes inflow quantities and negative outflow quantities. It is important to recognize the volumes of water that: (i) pass through lake Vrana bed (red areas), (ii) pass through Prosika channel bed (black areas), and (iii) overflow the weir crest at 0.41 m a.s.l. at the end of Prosika channel (orange areas). Namely, as the sea level in August 2011 and 2012 was above the lake water level ($h_s > h_l$), the model
predicts the intrusion of salt water through lake Vrana bed and also a small contribution from Prosika channel bed (red and black areas on the positive side). Moreover, the volume analysis can be used to estimate the amount of salt water in lake Vrana.

Under assumption of exclusively fresh water in lake Vrana at the beginning of simulated time period and the expectation that it is difficult for the contained salt water to mix with fresh water hence it is firstly squeezed out through groundwater flow if the required pressure gradient are reached (as it is denser and so close to the lake bed), the volume analysis can be used to
estimate the temporal change in the ratio of fresh and salt water in lake Vrana, thus giving useful information on water quality. Accordingly, Fig. 10 illustrates the change in the ratio between fresh and salt water in lake Vrana over the considered time domain. Moreover, the amount of salt water penetrated during the low lake water level can be monitored, as well as the gradual extrusion of salt water as a consequence of replenishment of fresh water in the coming period of floods.

### 4.3 Analysis of the tested protection against seawater intrusion

The effectiveness of different technical solutions for protection against excessive seawater intrusion can now be tested by changing the appropriate model parameters with the intention of simulating different flow conditions. In order to illustrate the application of the model, the quantity and quality of water in lake Vrana (related to lake water level and ration between fresh and salt water) was modeled adopting a tested protection against seawater intrusion involving: (i) the construction of a movable gate at the end of Prosika channel (instead of the current weir of fixed height), and (ii) lining of the canal to prevent seawater

intrusion through Prosika channel bed, denote by black areas in Fig. 9. Namely, a movable gate is necessary in order to preserve the possibility of evacuating a large amount of water during the flooding period. At the same time the movable gate can be used to prevent water overflow (denoted by orange areas in Fig. 9), thus achieving the retention of fresh water in the lake.

To illustrate the benefit of the introduced movable gate, it was necessary to define an algorithm for its movement during dry and wet periods. For this purpose, the gate movement was defined as a function of the current lake water level $h_l(t)$. In order

to contribute to the retention of fresh water during dry period, the gate crest in the lowered position was set at 1.05 m a.s.l. preventing overflowing of fresh water and saltwater intrusion by surface flow in Prosika channel (term $q_{cs,sw}$). During flood periods, the gate is raised so that the overflowing crest level is at 0.15 m a.s.l., i.e. at the Prosika channel bottom (thus achieving maximum throughput).

For the described gate control algorithm, Fig. 11 shows the achieved lake water level variation in the considered time domain

and under the same hydrological conditions as used previously. By comparing the obtained time change of lake water level (red line) with the one measured for the existing state of protection against seawater intrusion (blue line), it should be noted that the lake water level increases negligibly during the dry period in August 2012. However, the effectiveness of the considered protection can be recognized by performing the decomposition of water volumes entering or leaving lake Vrana per unit time (as in the previous case). Accordingly, Fig. 12 shows the decomposition of water volumes obtained for the tested protection

case. The lining of channel bed excluded the component of water exchange that takes place through porous Prosika channel bed, denoted by black areas in Fig. 9 obtained for the existing protection case. Notwithstanding, it should be noted that the presence of a gate increased the flow through lake Vrana bed, denoted by red regions in Fig. 12. This result should not be surprising, because the gate being active during most of the time raises the lake water level, as shown in Fig. 11, and consequently raises the pressure gradient and in turn the component of sinking flow described by the term $q_{ls,gw}$. The benefit of the tested protection

case can be recognized if the temporal change in the ratio of fresh and salt water is observed (as previously). Although the flow of water from the lake is higher (resulting from rising water levels in the lake), the total amount of salt water in the lake decreases as a consequence of lining Prosika channel bed (preventing a one component of seawater intrusion) and also retaining a larger amount of fresh water in the lake. The resulting benefit can be recognized by comparing Fig. 10 and Fig. 13. Finally, it should be also noted that the gate presence with the given maneuvering algorithm did not disrupt flood protection as

the maximal lake water level did not increase.

## 5 Conclusions

The application of a semi-distributed lumped karst model requires the estimation of hydraulic conductivity functions by which the relationship between the difference in pressure head along karst conduits (related to pressure gradient) and the achieved groundwater flow is described. An iterative method based on the application of PSO method has been proposed for the estimation of these functions. Accordingly, the inverse modeling task is considered as an optimization problem which is carried out by minimizing the objective function through which the difference between the predicted hydrological quantity and the measured one (e.g. water level under same hydrological conditions) is quantified. For this purpose, it was necessary to provide all relevant hydrological and other data in a representative time domain that includes hydrological extremes. For illustrative purposes, the proposed procedure was applied to model the exchange of fresh and salt water in lake Vrana in Croatia. To estimate the hydraulic conductivity functions related to groundwater flow between lake Vrana and surrounding karst aquifer and Adriatic sea, a time domain that spans over 6 years was considered. Apart from the unknown hydraulic conductivity functions, the functions that relate groundwater level in karst aquifer to corresponding horizontal-cross area of karst conduits where also estimated by the procedure. To reduce the number of all possible solutions, additional constraint conditions were applied to the unknown functions. Within an reasonable number of evaluation steps, the procedure converged to a solution by which the computed time change in the lake water level was approximately equal to the measured lake water level over the entire time domain (including dry and wet periods).

The calibrated model was used to analyze the current protection of lake Vrana from saltwater intrusion. For this purpose, a decomposition of the total water volume inside lake Vrana was conducted based on the origin of water and direction of the flow (inflow/outflow). By assuming no salt water at the beginning of the time domain, the conducted analysis was used to monitor the volume ratio of fresh and salt water in the lake. To increase the quality but also the quantity of water in lake Vrana, an alternative protection from saltwater intrusion was tested. The considered protection case included: (i) lining the Prosika channel bed, and (ii) construction of a movable gate at the end of Prosika channel. For the same hydrological conditions and time domain, the tested protection solution did not prevent the lake water level from falling to the lowest point (as for the existing protection system). However, the decomposition of water volumes that enters and exits the lake revealed a smaller amount of salt water compared to the existing protection system.

*Data availability.* All raw data can be provided by the corresponding authors upon request.

*Author contributions.* All authors equally contributed to the development of the presented methodology for estimating hydraulic conductivity functions and writing the paper. The conceptual model was developed by V. Travaš while the numerical model was developed by L. Zaharija and D. Stipanić. The framework of model calibration by stochastic optimization was designed by V. Travaš and S. Družeta. Data analysis and preparation of figures were performed by L. Zaharija and D. Stipanić. Pre-submission manuscript review was done by S. Družeta.

*Competing interests.* The authors declare that they have no conflicting interests.

*Acknowledgements.* This research article is a part of the project Computational fluid flow, flooding, and pollution propagation modeling in rivers and coastal marine waters–KLIMOD (grant no. KK.05.1.1.02.0017), and is funded by the Ministry of Environment and Energy of the Republic of Croatia and the European structural and investment funds.

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

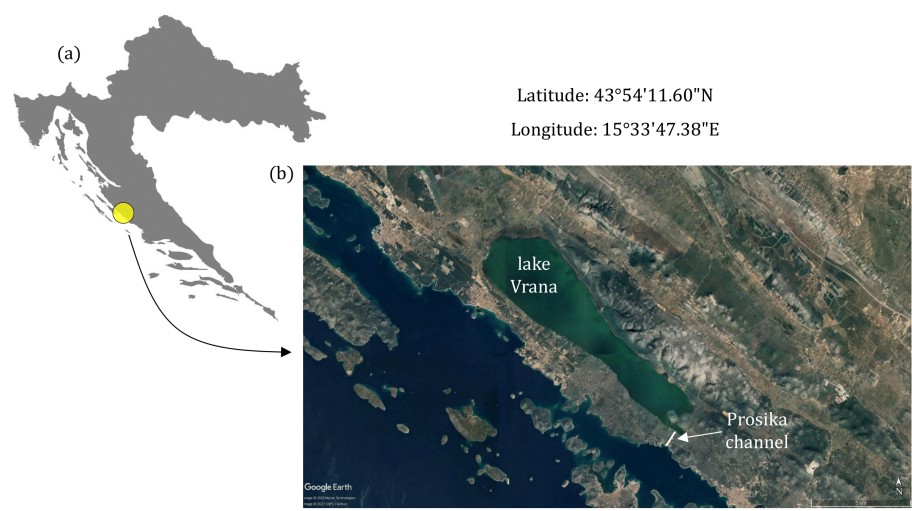

**Figure 1.** (a) Geographical position of lake Vrana, Croatia and (b) satellite view of the lake with the marked area of the Prosika channel through which the exchange of fresh and salt water is conducted by surface water flow (depending on the established boundary conditions). Maps Data: Google Earth, Image ©2022 Maxar Technologies, CNES / Airbus.

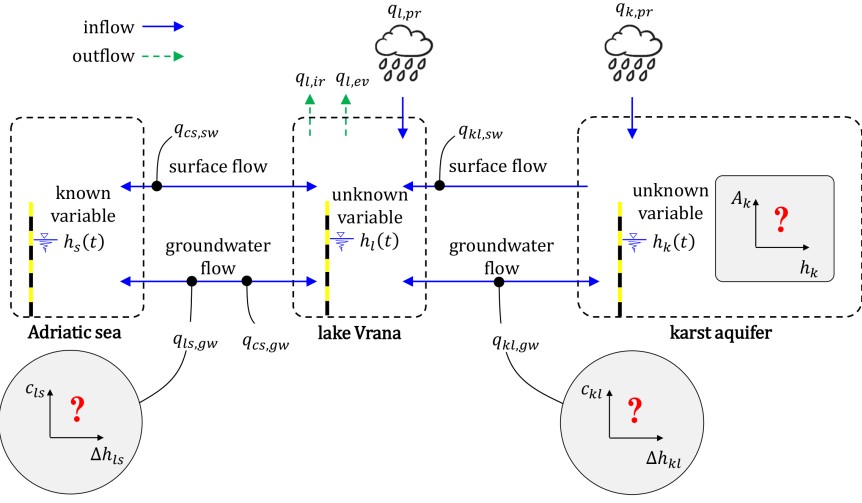

**Figure 2.** A conceptual model used to describe the storage dynamics of lake Vrana and its hydrogeological connectivity with: (i) Adriatic sea, and (ii) surrounding karst aquifer.

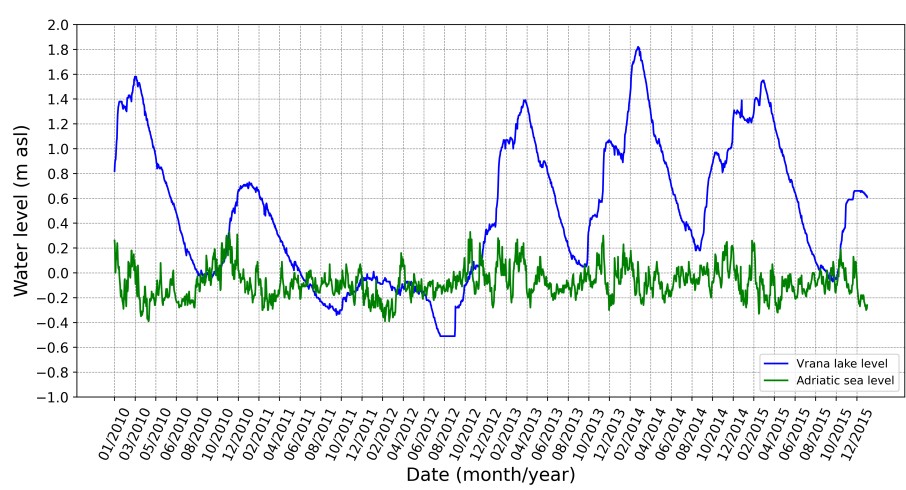

**Figure 3.** Measured lake water level and sea level in the time domain from early 2010 to late 2015. It should be noted that in August 2011 and 2012 the sea level was above the lake water level and so the intrusion of salt water was significant. In addition, the lake water levels after these events should be viewed in the context of the weir crest at 0.41 m a.s.l. identify overflow during flood waves.

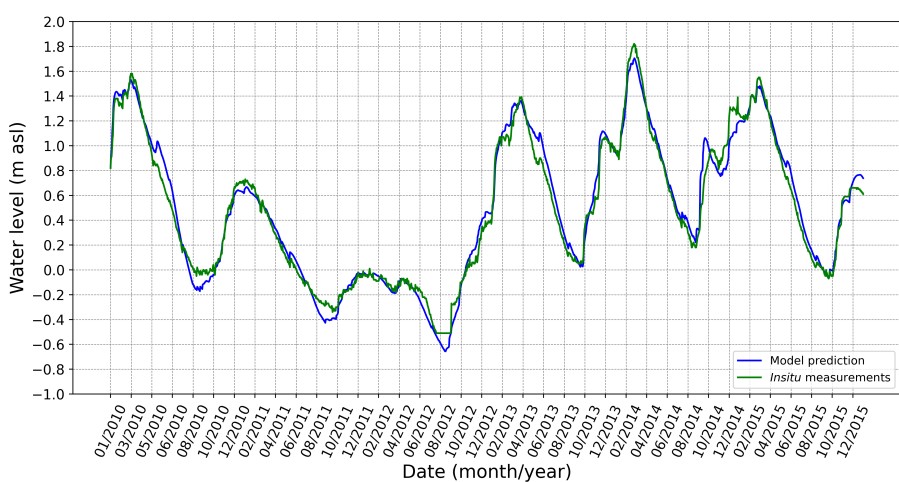

**Figure 4.** The predicted change in lake water level $h_l(t)$ (blue line), obtained by functions illustrated in Fig. 6, 7, and 8, and the change in lake water level $\hat{h}(t)$ determined by *in situ* measurements (green line). In the selected time domain, the drought period in August 2012 should be noted, as well as the next three extremes that arose as a result of flood waves.

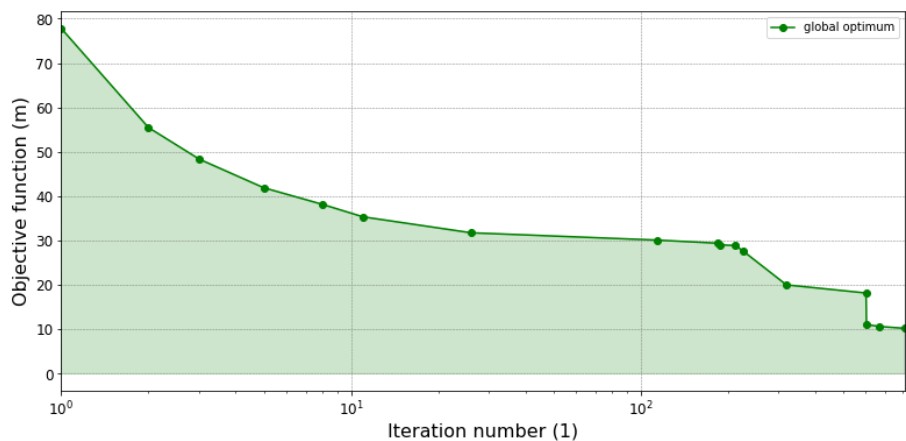

**Figure 5.** Global optimum points obtained for the design variables $\mathbf{x}_{g,best}^{(e)}$ by which the objective function reaches the so far established minimum.

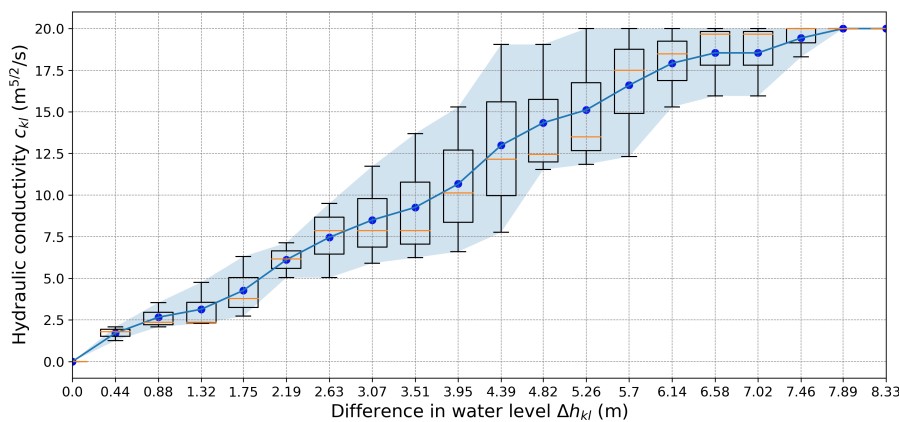

**Figure 6.** Statistical distribution of estimated values of the hydraulic conductivity function $c_{kl}(\Delta h_{kl})$ in (6) and (7) required for quantifying groundwater flow between karst aquifer and lake Vrana, obtained by model calibration over a 6 year time domain (from early 2010 to late 2015) by PSO method and adopting constraint conditions of increasing in function values with increasing the absolute values of pressure head difference $\Delta h_{kl}$. It should be noted that the iterative evaluation procedure of the design functions has converged into a solution that does not predicts the exchange of water from the direction of lake Vrana to the surrounding karst aquifer (which was to be expected and is recognizable in the function domain).

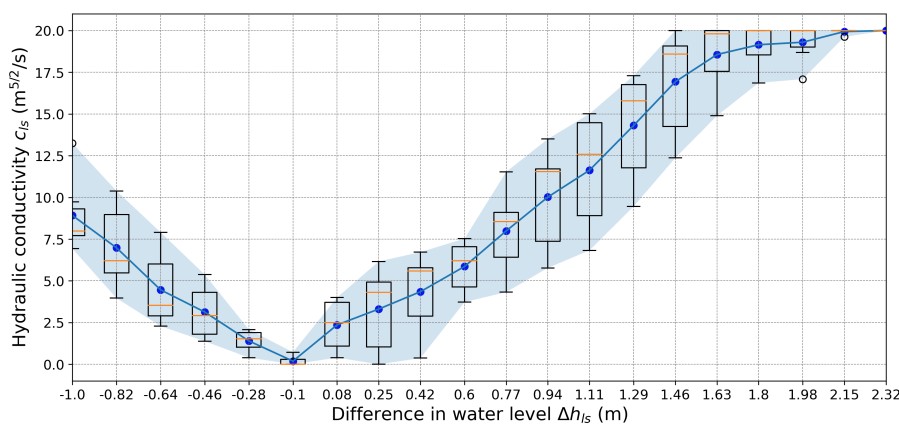

**Figure 7.** Statistical distribution of estimated values of the hydraulic conductivity function $c_{ls}\left(\Delta h_{ls}\right)$ in (7) required for quantifying ground-water flow between lake Vrana and Adriatic sea, obtained by model calibration over a 6 year time domain (from early 2010 to late 2015) by PSO method and adopting constraint conditions of increasing in function values with increasing the absolute values of pressure head difference $\Delta h_{ls}$.

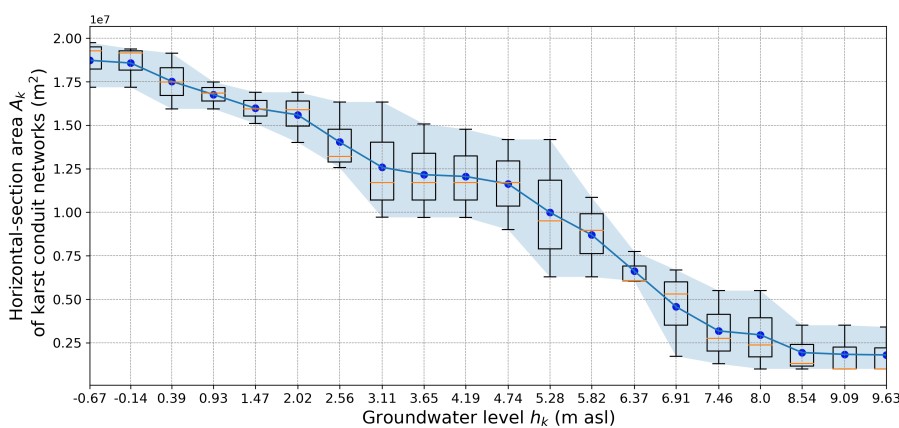

**Figure 8.** Statistical distribution of estimated values of the function $A_k(h_k)$ in (6) obtained by model calibration by PSO method over a 6 year time domain (from early 2010 to late 2015) and adopting the constraint condition of progressive reduction in function values i.e. requiring negative derivation at each calibrating point.

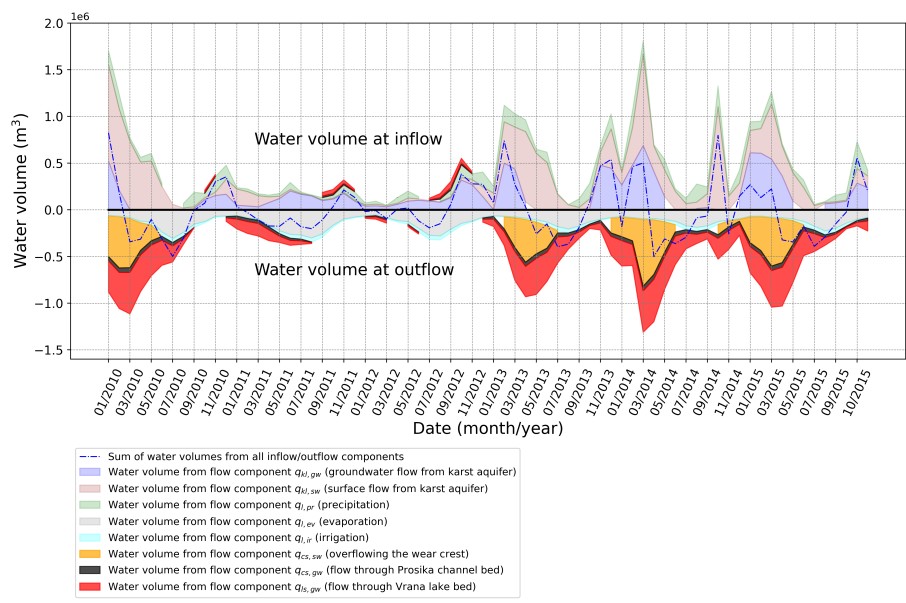

**Figure 9.** The decomposition of water volumes related to inflow and outflow from lake Vrana in the considered time domain and for the existing protection from seawater intrusion (positive values denotes volumes entering the lake and negative the opposite). The intrusion of salt water can be recognized through lake Vrana bed and Prosika channel bed.

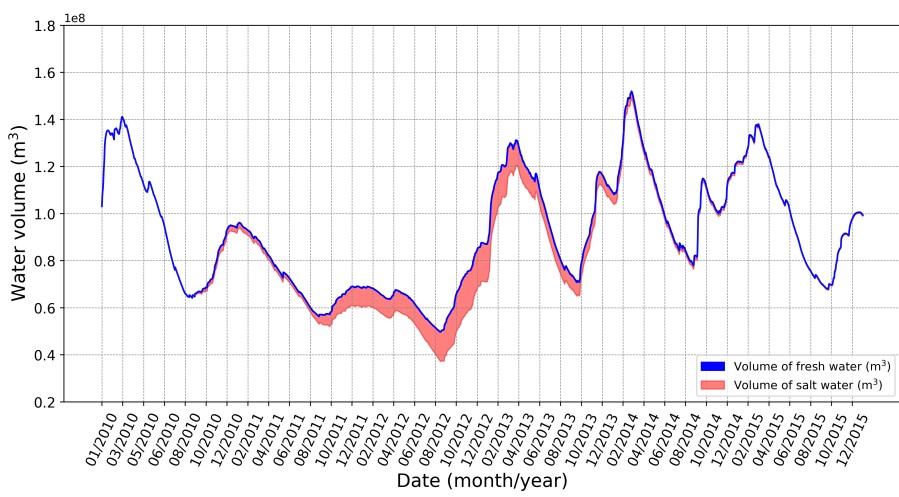

**Figure 10.** The temporal change in a ration between fresh and salt water in lake Vrana in the considered time domain and for the existing protection from seawater intrusion. It should be noted that the intrusion of salt water begins in September 2010 when in a short time period the lake water level falls below sea level as shown in Fig. 3.

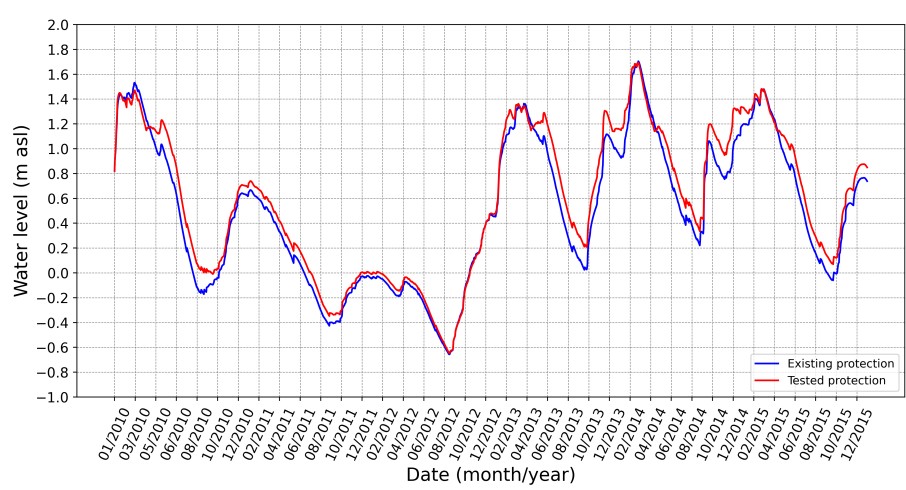

**Figure 11.** Comparison of the measured lake water level (blue line) for the existing protection from seawater intrusion and lake water level obtained for the tested protection from seawater intrusion (red line) which includes: (i) the moving gate at the end of Prosika channel, and (ii) lining the Prosika channel bed.

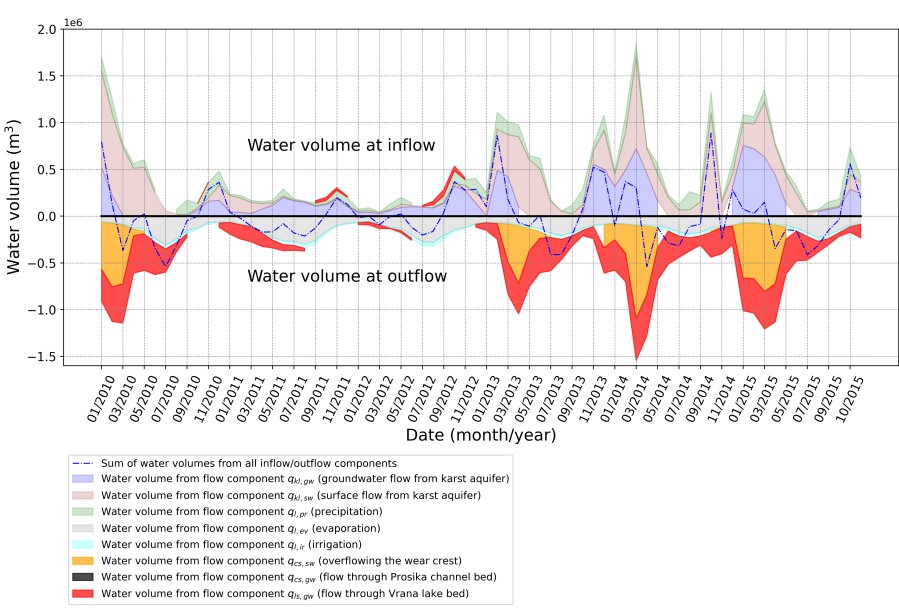

**Figure 12.** The decomposition of water volumes related to inflow and outflow from lake Vrana in the considered time domain and for a hypothetical protection variant involving lining the Prosika channel bed and raising the level of the downstream overflow.

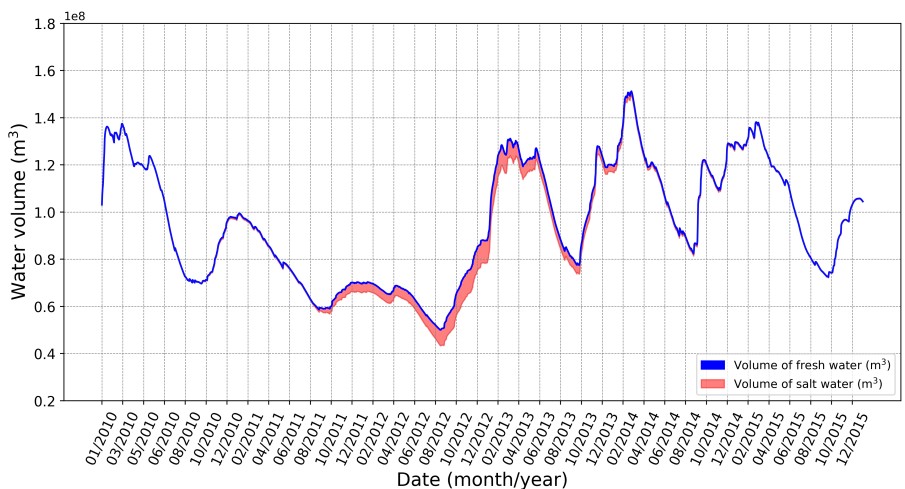

**Figure 13.** The temporal change of the ration between fresh and salt water in lake Vrana in the considered time domain and for a tested protection involving the moving gate and the end of Prosika channel and lining the Prosika channel bed.