# Peer review of "Estimation of hydraulic conductivity functions in karst regions by particle swarm optimization with application to lake Vrana, Croatia"

_Hydrology and Earth System Sciences, 2022_

## Author Response (AR2)

Dear Editor,
We are very pleased that the manuscript has been accepted for publication!

I would only ask you to notice that figure 5 has a slightly lower resolution (we did not notice this before) and that, if necessary, we can replace it with another figure of a higher resolution. We have added a variant of the figure with a higher resolution to the corresponding folder. Note the diagrams are slightly different because each analysis differs by the initial position of the PSO particles, which are initially given randomly (however, the convergence is the same).

A point-by-point response to the reviews is given hereafter.

Best regards,
Authors
* * *
**R1**

**REMARK**: This paper presents an iterative method based on the application of particle swarm optimization for estimating the hydraulic conductivity functions associated with the semi-distributed lumped karts model in lake Vrana, Croatia. In my opinion, the topics of this paper might be of interest to the readers of this journal, but it cannot be considered acceptable for publication in its current state. I suggest that the authors consider a major revision of their work along the following comments.

**ANSWER**: We would like to thank the reviewer for all the remarks and comments, and we hope that with the answers given below and the changes made in the paper we managed to achieve the required quality for publication. We responded to all the comments and suggestions, and based on them, we believe that we have made appropriate corrections in the paper, which are marked by red text. Please note that we have made corrections and adaptations in the paper based also on the remarks and suggestions of other reviewers and so some corrections can overlap.

**REMARK**: An important concern is that the paper does not appear to be significantly innovative, but demonstrates a complex exercise that applies some approaches well established in the literature. The novelty of this paper should be reinforced to illustrate the scientific and academic findings in the study.

**ANSWER**: Please note that the model in consideration is represented by a system of two nonlinear ordinary differential equations with variable coefficients (i.e. three calibration functions). Since it cannot be classified as an LTI system, it was to be expected that the task of calibrating such a model would be very complex (note that the calibration functions and also their domains are unknown). The calibration of the model first started with the trial-and-error method, and we immediately established that there is an exceptional sensitivity of the model results to even the smallest changes in the calibration functions. For this reason, we started trying other calibration methods, and of all the ones we tried, the PSO method was convincingly the most effective. This paper was written in order to share our experiences. In this sense, we believe that the modest novelty of the paper can be recognized in the application of the PSO method to the calibration of this type of model. Namely, by reviewing the literature, we did not find the same use of the PSO method. Moreover, we came across only a few papers that use the keywords PSO and karst model (but not in the same context as in this paper). In order to highlight the modest scientific significance of the paper, we supplemented the introduction by elaborating the problem of multimodality (that is encountered in such problems)

and brought it into connection with the application of the PSO method and the considered modeling approach by system of ODEs (i.e. lumped karst models).

**REMARK**: Another major concern relates to the estimate of the hydraulic conductivity functions of the karst aquifer. Although the mathematical model for simulating the exchange of fresh water and salt water is generally well calibrated, the uncertainty associated with precipitation recharge should be considered in model calibration. I strongly recommend authors investigate the sensitivity analysis regarding the inflow from precipitation recharge within the lake watershed.

**ANSWER**: We would like to recognize that the precipitation data are included in the groundwater flow component $q_{kl,gw}$ and the surface flow component $q_{kl,sf}$ which was obtained by field measurements (the same as the data of the flow component $q_{l,pr}$ from precipitation on the lake itself). In this way, these terms appear on the RHS as known functions in time and are not the subject of modeling. If the remark meant the model's sensitivity to these parameters, we would ask you to note that this was not the subject of the paper, but we focused on the problem of model calibration. Although we can equip the paper with a shorter sensitivity analysis, we believe that a complete sensitivity analysis would exceed the scope of this paper or would require writing a new paper. On the other hand, if the remark meant the sensitivity of the model with respect to the calibration functions, then we can state that the model result is extremely sensitive to a small change in the calibration functions (which was to be expected due to the nonlinearity of the model – note that the surface flow in Prosika channel is also modeled). For this very reason, it was necessary to apply the PSO method, which proved to be the best for global and local search of the domain of the objective function. The PSO method was used to search for the best solutions of the calibration functions, which will make the model solution the least sensitive to their change. Since the calibration functions are independent of time (model parameters), the obtained calibrated model represents the basis for further research, among which there will certainly be an analysis of the sensitivity of the model results to precipitations (we believe that this topic should be considered in another separate paper).

Fig. 7 should be explained in detail. Nowhere can be found the description of groundwater level in the domain of interest.

**ANSWER**: An additional description of the Figure in question is added in the paper (marked by red text). Namely, Fig. 7 shows the arrangement of the cross-section surface areas of cracks in the karst environment with respect to the elevation and thus includes the surfaces of conductors, caves, caverns, etc. Like hydraulic conductivity functions, this function is also unknown in advance and it is most often known that it decreases due to the dissolution process with the rise of the groundwater level (caverns and caves are usually located at deeper areas of the aquifer). On the other hand, as the groundwater level (variable $h_k$) was one of the variables in the considered system of ODE, at the beginning of the time domain it was set as an initial condition (like the variable $h_l$ used to represent the water level in the lake). A description has been added for the interpretation of Figure 7, as well as a text related to the initial condition of groundwater level. "The initial conditions were given by the model variables $h_l(t_0)$ and $h_k(t_0)$ defined at time $t_0$ i.e. at the beginning of the time domain. The initial condition $h_l(t_0)$ was set to 0.81 m a.s.l. which is known by field measurements (as can be recognized in Fig. 3). On the other hand, the variable $h_k(t_0)$ was set to 2.2 m a.s.l. and defined from model calibration so that a relatively rapid raise in water level $h_l$, at the beginning of the time domain, is obtain (as evidenced by in situ measurements shown in Fig. 3)."

**REMARK**: If there are some measurement data of water salinity over time available, it's better to consider the mass exchange of salinity between fresh water in karst aquifer and salt water in lake/sea coupled in the conceptual model.

**ANSWER:** Unfortunately, salinity measurements are only available for the lake and not in a continuous time series. On the other hand, it should be recognized that the lake is fed with fresh water from the karst aquifer, and therefore the flow in that part is only directed towards the lake (which is confirmed by the model i.e. sign of the flow component qgw,kl in Fig. 8 and Fig. 11). In this sense, there is no exchange of salt water between the lake and the karst aquifer.

**REMARK**: Some specific typos are below:

(1) Line 53, salt water instruction --> salt water intrusion

(2) Line 75, can by used to --> can be used to

(3) Line 312, is consider as --> is considered as

**ANSWER**: In accordance with the remarks, the necessary corrections were made.
* * *
**R2**

**REMARK**: The authors use a system of two ordinary and nonlinear differential equations to describe the exchange of fresh and saltwater between the lake and its surroundings. The method of particle swarm optimization was used to optimize the model. The authors show a complex exercise of the model in the literature.

**ANSWER**: Thank you for all the constructive comments and remarks. Allow us to address the same below with a few non-pretentious answers.

**REMARK**: However, the paper does not appear to be significantly innovative.

**ANSWER**: In order to highlight the scientific significance of the paper, we supplemented the introduction by elaborating the problem of multimodality encountered in such modeling tasks and brought it into relationship with the application of the PSO method and the considered modeling approach by system of ODEs (i.e. lumped karst models). In addition to the mentioned, the list of literature was expanded in order to bring this paper into relationship with previous papers in which the problem of optimization was considered (general optimization and optimization of hydrological models).

As you stated in the review, the model in consideration is represented by a system of two nonlinear ordinary differential equations with variable coefficients (i.e. three calibration functions). Since it cannot be classified as an LTI system, it was to be expected that the task of calibrating such a model would be very complex (note that the calibration functions but also their domains are unknown). The calibration of the model first started with the trial-and-error method, and we immediately established that there is an exceptional sensitivity of the model results to even the smallest changes in the calibration functions. For this reason, we started trying other calibration methods, and of all the ones we tried, the PSO method was convincingly the most effective (due to the multimodality of the problem and the need to perform a global and local search of the target function domain). This paper was written in order to share our experiences. In this sense, we believe that the modest novelty of the paper can be recognized in the application of the PSO method to the calibration of this type of karst models. Namely, by reviewing the literature, we did not find the same use of the PSO method.

Moreover, we came across only a few papers that use the keywords PSO and karst model (but not in the same contest as in this paper).

**REMARK**: It is suggested that the precipitation recharge should be considered in the model, which could influence the model obviously.

**ANSWER**: In order to be able to implement appropriate changes or respond to your remarks, we would kindly ask you to clarify the statement "precipitation recharge should be considered". Thank you! We would like to recognize that the precipitation data are included in the groundwater flow component $q_{kl,gw}$ and the surface flow component $q_{kl,sf}$ which was obtained by field measurements (the same as the data of the flow component $q_{l,pr}$ from precipitation on the lake itself). In this way, these terms appear on the RHS as known functions in time and are not the subject of modeling. If the remark meant the model's sensitivity to these parameters, we would ask you to note that this was not the subject of the paper, but we focused on the problem of model calibration. Although we can equip the paper with a shorter sensitivity analysis, we believe that a complete sensitivity analysis would exceed the scope of this paper or would require writing a new paper. On the other hand, if the remark meant the sensitivity of the model with respect to the calibration functions, then we can state that the model result is extremely sensitive to a small change in the calibration functions (which was to be expected due to the nonlinearity of the model – note that the surface flow in Prosika channel is also modeled by not explained in details in the paper). For this very reason, it was necessary to apply the PSO method, which proved to be the best for global and local search of the domain of the objective function. The PSO method was used to search for the best solutions of the calibration functions, which will make the model solution the least sensitive to their change. Since the calibration functions are independent of time (model parameters), the obtained calibrated model represents the basis for further research, among which there will certainly be an analysis of the sensitivity of the model results to precipitations (we believe that this topic should be considered in another separate paper).
* * *
**R3**

**REMARK**: This manuscript estimated the hydraulic conductivity using particle swarm optimization method and analyzed the effect of protection method for the lake Vrana. The topic is somehow interesting, however, I have some major concerns for this manuscript as shown below:

**ANSWER**: We would like to thank the reviewer for all the remarks and comments, and we hope that with the answers given below and the changes made in the paper we managed to achieve the required quality for publication. We responded to all the comments and suggestions, and based on them, we believe that we have made appropriate corrections in the paper, which are marked with blue text. Please note that we have made corrections and adaptations in the paper based also on the remarks and suggestions of other reviewers and so some corrections can overlap.

**REMARK**: The introduction is not well organized. A literature review for the optimization method for the hydrological model and groundwater flow model is missing. Therefore, why do you select the particle swarm optimization method and not select other methods? I suggest to add an introduction for the literature review for the method of optimization of model parameters.

**ANSWER**: We have reorganized the introduction. In order to explain the issue of calibrating hydrological and groundwater models in more detail, we have further elaborated and enlarged the introduction and accordingly increased the list of literature (the last paragraph of the introduction has been completely replaced and significantly expanded). Apart that, in the introduction we also

stated the problem of multimodality and used it as an argument for the application of the PSO method. Other organizational corrections were made in the introduction (all marked in the new version of the paper). A new part of the introduction (the last paragraph of the introduction), as well as the newly introduced references, are listed below.

\bibitem[Beasley et al. (2012)]{Beasley1993}
Beasley, D., Bull, D.R., and Martin, R.R.: A Sequential Niche Technique for Multimodal Function Optimization, Evolutionary Computation, 1(2), 101-125, doi: 10.1162/evco.1993.1.2.101., 1993.

\bibitem[Beven (2006)]{Beven2005}
Beven, K.J.: A manifesto for the equifinality thesis, J. Hydrol., 320(1-2), 18-36, 2006.

[revised manuscript text omitted]

**REMARK**: The assumption of fully turbulent and partially saturated water flow through karst conduits is used and the Darcy's flow is neglected for the groundwater flow simulation. Does this assumption cause uncertainty for the simulation? The reasonability for this assumption need further explanation. The hydrogeology conditions for the study area need some more detail introduction.

**ANSWER**: The introduced assumption certainly causes uncertainty, but with negligible influence for the scope of the model. Namely, the assumption can be justified by the purpose of the model, which was to describe the mass exchange of salt and fresh water between the lake and the sea. For this purpose, it should be noted that the lake is located in the immediate vicinity of the sea and that even a relatively small pressure gradients are reflected in the rapid exchange of water through karst conduits (which was also determined by field measurements in cases of different sea and water levels in the lake). The Darcy's flow cannot compete in the rate of exchange of these quantities of water and is therefore justifiably neglected. In the same time, in the used framework of lumped karst models the Darcy flow component is not naturally included and combined modeling requires the application of the dual porosity model which cannot be linked to the subject of this paper which is devoted to the application of PSO methods for calibration of hydraulic conductivity functions. The hydrogeology conditions of the study area are further elaborated in the title Study area where the text below is added.

Within the basin of lake Vrana, few groups of rocks can be recognized \citep{Rubinic2014}. First of all, these are Upper Cretaceous limestones, i.e. very permeable rocks within which an underground hydrographic network has been developed. On the other hand, it is also possible to determine the area within which the dolomites and limestones of the lower part of the Upper Cretaceous alternate, forming a medium permeable layer that can slow down the flow of underground water. Finally, a large part of the basin consists of impermeable or very poorly impermeable flysch deposits that in some places cause the formation of surface flows. For calibrating the model parameters, these surface flow components will be set based on known \textit{in situ} measurements. On the other hand, the groundwater flow components, which are realized as a consequence of the developed hydrographic network in the Upper Cretaceous limestones, will be modeled using the semi-distributed lumped karst model, relying on the assumption of a fully turbulent flow.

**REMARK**: For the parameter optimization, it is necessary to show the process for the seeking the optimal parameters, and how about the efficiency of this method? I suggest to add some comparison of this method and other traditional method.

**ANSWER**: To illustrate the convergence characteristics of the optimization procedure, a Figure has been added in the paper that shows the change in the global optimum during iterations (the best values of the objective function up to that iteration). It should be noted that complete convergence with a negligible error is not possible for the reason that the aforementioned would require a more detailed parameterization of the calibration functions (with more than 60 points – this comment is also important and is introduced in this new part of the paper). On the other hand, a more detailed parameterization of the calibration functions would make the calibration problem far more complex and would drastically affect the efficiency of the aforementioned method. Namely, in that case, the search space would be much larger, i.e. with a much larger number of dimensions. The efficiency of the method decreases drastically with the increase in the number of optimization variables, but at the same time it should be noted that it is subject to parallelization and that within one iteration the searching procedure of each particle is independent all other particles (which is attractive for

openMP parallelization). Regarding the comparison of this method with other similar methods, it should be noted that for the given framework (lumped karst model) only automated calibration methods can compete (neural networks, deep search, genetic algorithms, etc.). On the other hand, although there are essential differences between the above, the PSO method is particularly attractive for the reason that it offers a simple implementation of constraints conditions over calibration parameters and for the reason that it performs global and local search simultaneously (which solves the problem of multimodality). For the description of the convergence, we have added the text attached below.

The convergence of the optimization process is illustrated in figure \ref{fig08} which shows the value of the objective function at points $\textbf{x}_{g,best}^{(e)}$ of the global optimum respect to the iteration number. For the adopted parametrization of the calibration functions, the objective function reached the lowest possible value, and further reduction of its value would require a larger number of parameters, i.e. a denser discretization of the calibration functions (more than 20 point per functions). On the other hand, such a procedure would significantly affect the number of necessary iterations to reach a smaller error, as well as the number of required particles (because the search space would be larger). In this sense, the parameterization of the calibration functions is determined based on a compromise between computational time and acceptable minimum value of the objective function.

**REMARK**: A discussion for the comparison of results and method of this study and other similar studies are need.

**ANSWER**: For these purposes, we have added the text below.

In order to compare the presented approach with other approaches, it should be emphasized that the framework of the model is defined by a system of two ordinary and non-linear differential equations with variable coefficients that also define the calibration functions. In these circumstances, automated methods such as genetic algorithms are usually used \citep{Lu2011,Nematolahi2018}. On the other hand, if by the method of trial and error it is determined that the model results are significantly sensitive to the calibration functions (model parameters), the application of genetic algorithms is probably not the most appropriate. Namely, this sensitivity of model results to calibration functions indicates a large number of local minima of the objective function by which the multimodality of the problem can be recognized. In such circumstances, it is not only necessary to carry out a global search of the domain of the objective function, as carried out by the method of genetic algorithms, but it is also necessary to examine local minima, that is, to enable a more detailed search of individual parts of the domain by carrying out local searches. Moreover, the search for local minima must be adaptive so that the solution in the current iteration can be updated by the new local solution that results in a more favorable variant of the calibration parameters. In this way, the possibility of searching for a larger number of local minima is realized, which is necessary for such multimodal problems. Considering all of the mentioned, and by noting that the model in question showed the characteristics of multimodal problems, the calibration of the model was performed using the PSO method which simultaneously performs global and local search of the domain of the objective function \citep{Ozcan2007,Kuok2012,Zambrano2013}. Considering the experience gained from the performed analysis, the application of the PSO method can be recommended for the calibration of a semi-distributed lumped karst models based on a system of nonlinear ODEs.

**REMARK**: The conclusion seems too long, you can reduce some statements.

**ANSWER**: The conclusion is shortened.

Dear Editor,

Thank you for the instructions given to improve the quality of the manuscript. In addition to the previous changes in the manuscript, which we made based on previously submitted reviews, we also made new changes that were motivated by your last review of the manuscript. Therefore, we took into account the uncertainty in the precipitation data and examined their influence on the model calibration procedure (sensitivity analysis). For this purpose, we needed a little more time because we had to implement certain adaptations in the code and performed an additional 200 different model calibrations (we needed parallel programming to make this possible). Accordingly, the results of the calibration functions are now presented using box diagrams (statistical distribution of function values for each point of the domain). As the reviewers correctly predicted, it turned out that the uncertainty in precipitation is significant for the outcome of the calibration. At the same time, it is interesting to note that the significance of the uncertainty of the input data varies for individual segments of the domain of these functions, so the spread of possible values of the calibration functions is greater in some parts and less in some other parts of their domains (as shown in new figures). With these last adaptations in the code, the calibration procedure of the model became even more demanding computationally and more complex to implement in a computer code, but it made possible to identified the areas of relative water level differences that require additional in-situ research so that new conditions for the optimization of unknown functions could be imposed in calibration procedures. In any case, we think that we have shown that the application of the PSO method is very applicable for lumped semi-distributed karst models and we have systematically presented the calibration procedure of a very demanding karst system. We hope that with these last corrections, we have reached the criteria for publication.

As requested, we have also elaborated the innovative contribution of the study. For this purpose, it should be noted that lake Vrana has not yet been modeled in this way, mainly because the model in question (defined by a nonlinear system of ODE) was demanding to calibrate. With this paper, we wanted to show the results of exhaustive modeling activities that made it possible to model this karst area successfully (judging by the model calibration i.e. judging by the comparison of the modeled and measured water level in the lake over a time span of 6 years with a time increment of 1 hour). Moreover, this model is also useful for defining an adequate solution for lake protection (which is also commented on in the paper). By reviewing other available papers, we did not come across the same or similar application of the PSO method for the purpose of calibrating unknown functions used in lumped semi-distributed karst models. On the other hand, by taking into account the influence of uncertainty in the input data, we have integrated a complete calibration approach that can be applied in an analogous way for other karst systems described by systems of nonlinear differential equations. As we stated in the manuscript, the significance of this approach can be recognized by comparing calibration attempts of similar models with other calibration methods. Namely, the complexity of the model in this case requires a simultaneous global and local search of the search space of all possible solutions, for which the PSO method proved to be successful.

Finally, we would like to note that the manuscript has been significantly enriched in the text and in the bibliography, and I would also like to state that we have corrected other minor mistakes that we missed in the previous version of the manuscript. Parts of the manuscript that have been modified are marked with colored text (red and blue text) as indicated in the previous round of review (each color for one reviewer).

We are at your disposal for any additional questions.

Best regards,
Authors.